# The Application of Heat-Shrinkable Fibers and Internal Curing Aggregates in the Field of Crack Resistance of High-Strength Marine Structural Mass Concrete: A Review and Prospects

**DOI:** 10.3390/polym15193884

**Published:** 2023-09-26

**Authors:** Jinhui Li, Zi Yu, Jing Wu, Qingjun Ding, Wei Xu, Shaolong Huang

**Affiliations:** 1School of Materials Science and Engineering, Wuhan Textile University, Wuhan 430200, China; yu1018541283@163.com (Z.Y.); wujing313@whut.edu.cn (J.W.); 2State Key Laboratory of Silicate Materials for Architectures, Wuhan University of Technology, Wuhan 430070, China; dingqj@whut.edu.cn; 3School of Engineering, China University of Geosciences, Wuhan 430074, China; 13871526941@139.com; 4School of Materials Science and Engineering, Hubei University, Wuhan 430062, China; huangsl@hubu.edu.cn

**Keywords:** high-performance concrete, heat-shrinkable fibers, internal curing, crack control, microstructure regulation

## Abstract

High-strength large-volume marine concrete is a critical material required for the construction of large-span sea-crossing bridges. However, the widespread issue of cracking in this concrete type significantly impacts the durability and load-bearing capacity of concrete structures. Dealing with these cracks not only delays construction schedules but also increases project costs. Addressing these pressing technical issues, this project proposes the use of newly developed high-modulus heat-shrinkable fibers (polyethylene terephthalate fiber, also known as PET fiber) from the textile industry. These fibers utilize the heat generated during the hydration of large-volume concrete to trigger its contraction, applying three-dimensional micro-prestressing stress to enhance its crack resistance, while simultaneously incorporating prewetted aggregates with high-performance micro-porous structures and utilizing their internal curing effect to reduce concrete shrinkage. This helps to minimize the loss of micro-prestressing stress caused by concrete shrinkage and creep. This synergistic approach aims to improve the crack resistance of high-strength large-volume marine concrete. By employing modern testing and simulation analysis techniques, this study aims to uncover the mechanism by which the heat-shrinkable fibers exert micro-prestressing stress on concrete and the water release mechanism of internal curing aggregates during the temperature rise and fall stages of large-volume concrete. It seeks to elucidate the cooperative regulation of the microstructure and performance enhancement mechanisms of high-strength large-volume marine concrete by the heat-shrinkable fibers and internal curing aggregates. This research will lead to the development of novel methods for the design and crack control of high-strength large-volume marine concrete, which will be validated through engineering demonstrations. The outcomes of this study will provide theoretical foundations and technical support for the preparation of the crack-resistant large-volume marine concrete used in large-span bridges.

## 1. Introduction

With the advancement of the “Belt and Road Initiative,” the strategic development of becoming a maritime power, and the progress in the development plan for the Greater Bay Area of Guangdong, Hong Kong, and Macau, numerous large-span sea-crossing bridge projects are either under construction or about to commence (Figure 1). In addition to utilizing medium-to-low strength grade large-volume marine concrete, such as C30 to C40, these projects necessitate the application of high-strength large-volume marine concrete exceeding C50 in critical structural elements (e.g., solid sections of C60 bridge towers, continuous rigid pre-stressed box girders ranging from C60 to C70, primary arch rings of C80 arch bridges, and combined sections of C100 steel and concrete), with an increasingly expansive adoption [1].

These extensive sea-crossing bridge projects not only provide support for the development of transportation infrastructure but also play a pivotal role in driving regional economic growth and cooperation. The application of high-strength large-volume marine concrete ensures the load-bearing capacity and durability of these bridges, enabling them to address the challenges posed by the maritime environment, such as salt spray, seawater erosion, and climate change. The design and construction of these projects necessitate a comprehensive consideration of material performance, structural stability, and environmental impacts. Amidst this developmental trend, domestic research institutions and construction companies continually strive to innovate materials, technologies, and methodologies to meet the demands of these large-volume marine engineering projects. These efforts not only contribute to the successful completion of engineering endeavors but also hold the potential to significantly enhance the technological prowess of domestic engineering construction. The construction of these expansive sea-crossing bridges will further facilitate interconnectivity between regions, bolster international collaboration, and pave new pathways for economic development.

The practical results of engineering endeavors indicate that compared with the commonly used large-volume C30 to C40 concrete in bridge projects, the design requirements for high-strength marine mass concrete exceeding C50 entail higher strength and enhanced resistance to chloride ion penetration. This necessitates a lower water–cement ratio and a higher amount of cement and binder materials. The greater heat generated during hydration accumulates, even when employing measures such as low-heat cement, substantial mineral admixtures, the circulation of cooling water, and controlling the temperature during casting. Consequently, the internal maximum temperature can still reach 70 to 90 °C [2,3,4], with a temperature differential exceeding 25 °C between the interior and surface of the concrete. This inconsistency in volume deformation leads to significant temperature-induced stress. Furthermore, the increased amount of binder materials and lower water–cement ratio result in greater concrete shrinkage, thereby generating substantial shrinkage stress. The combination of temperature-induced stress and shrinkage stress leads to widespread cracking in high-strength marine mass concrete (Figure 2a,b). Moreover, certain structural elements possess variable cross sections, dense reinforcement layouts, and require prestressing, making it impractical to install cooling water pipes for temperature reduction (as depicted in Figure 2c) and thereby exacerbating the occurrence of cracks.

After the cracking of large-volume marine concrete, chloride ions in seawater readily penetrate through the cracks and reach the reinforcing steel, resulting in steel corrosion and concrete structural damage and thereby compromising the load-bearing capacity and durability of the concrete structure [5,6,7]. The need for crack treatment not only delays project completion and increases bridge construction costs but also impacts the visual quality of the concrete. Therefore, it is crucial to research new methods for crack control in high-strength large-volume marine concrete, elucidate the mechanisms for microstructure regulation, and uncover the mechanisms behind the performance enhancement. This research holds significant importance in addressing the cracking issues and improving the durability of high-strength large-volume marine concrete. Furthermore, the damage caused by chloride-induced corrosion not only compromises structural integrity but also impacts the long-term usability of marine concrete structures. Traditional crack repair methods often involve labor-intensive processes and the use of external materials, which may not always provide comprehensive and sustainable solutions. Therefore, there is a growing need for innovative technologies that can effectively prevent or mitigate cracking issues while maintaining the mechanical performance and durability of large-volume marine concrete.

## 2. Research and Applications of Large-Volume Marine Concrete

Large-volume marine concrete is a critical material for the construction of marine bridge projects. The key to enhancing its performance lies in addressing the issues of cracking and chloride salt erosion. For commonly used C30 to C40 large-volume marine concrete, measures to improve crack resistance and chloride salt resistance include optimizing the mix design through the use of low-heat cement or reducing the cement content, incorporating high-volume mineral admixtures, and adding retarding agents to prepare low-heat hydration temperature rise large-volume concrete [8,9,10,11]. Additional crack-resistant functional components such as hydration heat inhibitors [12,13,14,15] and expansive agents [16,17] are also incorporated. Temperature control measures, such as the installation of cooling water pipes [18], raw material cooling, enhanced insulation, and the implementation of post-casting thermal control measures, are also employed to enhance the crack resistance of the concrete. Furthermore, in addition to the application of concrete surface anti-corrosive coatings and silane treatments, the incorporation of rust inhibitors [19], including organic, inorganic, migratory, and volatile corrosion inhibitors, as well as ion transmission inhibitors, is essential for significantly enhancing concrete’s resistance to chloride salt erosion, effectively prolonging the lifespan of structures, reducing maintenance costs, ensuring safety, minimizing environmental impact, and fortifying critical infrastructure. For high-strength large-volume marine concrete exceeding C50, a higher binder content and lower water–cement ratio are required. C50 concrete also involves the use of ultrafine mineral admixtures such as silica fume to meet the mechanical performance requirements [20] and achieve a denser pore structure, resulting in excellent resistance to chloride salt erosion. However, the higher binder content and lower water–cement ratio lead to increased hydration heat and significant shrinkage, making the concrete prone to cracking. Cracks in this concrete can subsequently reduce its resistance to chloride salt erosion, posing a challenge in terms of crack control that needs urgent attention.

During the construction of large-volume marine concrete [21,22,23,24], there is a significant risk of leakage in the cooling water pipes. Once leakage occurs, it becomes a pathway for chloride ion intrusion, which adversely affects the durability of the concrete structure. Additionally, for certain high-strength large-volume marine concrete structures, due to the constraints of dense reinforcement, pre-stressing, and embedded components, it is not feasible to install cooling water pipes (as shown in Figure 2c). Consequently, the risk of cracking is further exacerbated. Therefore, for high-strength large-volume marine concrete structures [25,26,27,28,29,30] above C50 where it is not possible to install cooling water pipes, further research is required to develop novel methods for controlling cracks.

Engineering practice has demonstrated that the addition of hydration heat suppressants in large-volume concrete can reduce the temperature rise caused by hydration heat and delay the occurrence of temperature peaks, resulting in a decrease of up to 5–8 °C in the maximum concrete temperature. For C30~C40 large-volume marine concrete, the inclusion of hydration heat suppressants can effectively control cracking when cooling water pipes are not installed [7]. However, for high-strength pre-stressed large-volume marine concrete above C50 where it is not feasible to install cooling water pipes, the internal temperature can reach 75–95 °C and the cooling effect of hydration heat suppressants is limited. The internal temperature remains relatively high, ranging from 70 to 90 °C, with significant temperature differentials between the interior and exterior, posing a significant risk of cracking due to temperature-induced stress. Furthermore, high-strength large-volume marine concrete is often used in pre-stressed structures and, due to the construction schedule requirements, pre-stressing is typically performed at 7 days. The inclusion of hydration heat suppressants may reduce early-age strength and affect the implementation of pre-stressing operations. Therefore, for high-strength pre-stressed large-volume marine concrete structures above C50 where it is not possible to install cooling water pipes, the development of new crack control materials is necessary.

Furthermore, for high-strength large-volume marine concrete above C50, the low water-to-cement ratio and the inclusion of fine reactive powder materials such as silica fume result in significant shrinkage. On the other hand, the addition of expansive agents competes for limited water [8] with cementitious materials, triggering expansion reactions. Additionally, the hydration rate of expansive agents is closely related to the temperature [9], and in large-volume concrete, the central temperature is higher, with significant temperature differentials between the interior and exterior. In the absence of cooling water pipes and without the inclusion of hydration heat suppressants, expansive agents exhibit a faster hydration rate in the internal regions of concrete, whereas the hydration rate is slower on the surface. This discrepancy increases the self-restraint of the concrete, thereby raising the risk of cracking. Research indicates that a combination of expansive agents and hydration heat suppressants [6] can reduce the temperature differential on the concrete surface, mitigating the adverse effects of expansive agents on crack resistance in large-volume concrete. However, for high-strength pre-stressed large-volume marine concrete structures above C50 where the requirements of pre-stressing operations limit the inclusion of hydration heat suppressants, the sole inclusion of expansive agents cannot meet the crack control requirements of the high-strength large-volume marine concrete. Consequently, there is a need to develop new materials with shrinkage control functionality.

With the increasing span of bridges, researchers have proposed the use of steel–concrete composite structures to address the crack resistance issues of large-volume concrete. In non-marine environments, the use of steel shells or steel tubes filled with expansive large-volume concrete can effectively address the challenges of crack control. However, in marine environments, particularly in splash or tidal zones, the steel shells or steel tubes are at significant risk of corrosion. Therefore, for the application of steel–concrete composite structures in high-strength large-volume marine concrete under marine conditions, it is imperative to address the long-term corrosion protection of the steel shells or steel tubes.

The process of marine erosion is a complex interplay of multiple factors. It triggers a cascade of profound physicochemical responses within the matrix of cementitious and slag-based slurries, as depicted in Figure 3 and Figure 4. Primarily, ions such as chloride and sulfate present in seawater engage in ion exchange and complexation reactions with constituents within the cementitious and slag-based slurries, instigating the partial dissolution of the cementitious minerals. Chloride ions interact with calcium aluminosilicates and hypochlorite, resulting in the formation of soluble chlorides that lead to the dissolution of crucial components such as the calcium aluminosilicates and hastening the loosening and deterioration of the concrete’s microstructure.

Furthermore, sulfate ions in seawater can react with compounds such as calcium aluminosilicates within the cementitious matrix, forming sulfate precipitates. The generation of sulfates triggers the degradation of calcium aluminosilicates, causing volumetric expansion and diminished strength in the slurry and thus severely compromising its mechanical properties and stability. Moreover, oceanic elements such as waves, tides, and currents instigate physical impacts and abrasion upon the cementitious and slag-based slurries. The force of waves and the abrasive action of particles induce the detachment and erosion of particles from the surface of the slurry, consequently affecting its appearance and mechanical attributes.

Lastly, the salts and alkaline substances present in seawater may incite alkali–aggregate reactions within the cementitious and slag-based slurries. These reactions lead to internal expansion and contraction phenomena, engendering micro-volume damage that diminishes the mechanical integrity and stability of the slurry. On the whole, the impact of marine erosion on cementitious and slag-based slurries encompasses a spectrum ranging from chemical reactions to physical degradation. To mitigate these effects, measures such as using materials with resistance to chloride intrusion, incorporating corrosion-resistant additives, enhancing density, and implementing protective coatings should be adopted. These measures can ensure the performance and reliability of cementitious and slag-based slurries in marine environments. In practical engineering applications, it is imperative to comprehensively consider these factors when formulating tailored strategies for material selection and structural design. This approach ensures the long-term stability and sustainable operation of structures.

## 3. The Impact of Fibers on the Performance of Large-Volume Concrete in Marine Engineering

Foreign researchers were the first to introduce polypropylene fibers and steel fibers into large-volume concrete to enhance its crack resistance. Polypropylene fibers effectively improve the toughness of concrete and significantly reduce the cracks caused by plastic shrinkage. However, they have lower modulus, poor dispersion, and can affect the pumpability of concrete. Steel fibers have higher elasticity and tensile strength. When added to large-volume concrete, they can increase the tensile strength by approximately 15% to 20%, thereby enhancing its crack resistance. However, conventional steel fibers carry the risk of corrosion in marine engineering and also increase construction costs. Furthermore, both the fibers currently used and the concrete itself are materials that undergo thermal expansion and contraction. When the concrete undergoes temperature rise and expansion, the fibers expand accordingly. Similarly, when the concrete cools down and contracts, the fibers also contract. Taking steel fibers as an example, they have a higher thermal expansion coefficient than concrete. During the temperature rise phase of large-volume concrete, the disparate thermal expansion coefficients between steel fibers and the cementitious paste in the concrete induce tension at the paste–fiber interface, resulting in the formation of microscopic cracks. The contraction of fibers during the cooling phase further promotes crack development (Figure 5). Therefore, although steel fibers can improve the tensile strength of concrete and prevent crack propagation, there is room for improvement in controlling the formation of cracks during the temperature rise and cooling phases of large-volume concrete. In conclusion, it is necessary to modify the aforementioned thermal expansion and contraction of fibers to enhance their crack resistance effect in concrete.

In recent years, the textile industry has developed a novel type of fiber that is distinct from the aforementioned heat-expansion fibers. Known as thermal shrinkage fibers, these fibers exhibit a contraction effect with an increase in temperature. They possess remarkable features, such as high modulus, adjustable shrinkage rate, a wide range of thermal contraction temperatures, high strength, chemical corrosion resistance, and cost effectiveness. Among these fibers is PET fiber, which is derived from the polymerization of polyethylene terephthalate. The specific parameters are detailed in Table 1. This particular polymer has a notably high melting point and melting temperature. Consequently, under conventional conditions, PET fibers require elevated temperatures to induce significant thermal shrinkage, thereby effectively offsetting the thermal expansion that occurs during the concrete hydration process and providing excellent preventative measures against the cracks induced by the thermal expansion in concrete. The specific manufacturing process of PET fibers involves the initial polymerization reaction of terephthalic acid and ethylene glycol, leading to the formation of polyethylene terephthalate (PET) polymer. Subsequently, the polymer is processed into PET fiber raw materials through melting and extrusion. Then, the PET raw materials undergo a spinning process in which the melted polymer is extruded through microplates or spinnerets to form continuous fibers. Finally, the resulting PET fibers are solidified and subjected to post-processing to obtain the finished product. The thermal shrinkage process of PET fibers corresponds to the heat-setting stage in fiber formation, during which the intermolecular bonding and strength of the fiber are enhanced, thus improving its stability. Furthermore, PET fibers exhibit minimal changes in size and performance as the ambient temperature rises or falls.

The utilization of heat-shrinkable fibers within voluminous high-strength marine concrete finds its purpose in meeting the demands of extreme engineering requisites intrinsic to maritime environments in an endeavor to heighten the durability, crack resistance, and overall performance of concrete structures. The intricate nature of marine surroundings renders high-strength concrete susceptible to minute crack propagation arising from challenges such as chloride ion erosion and thermal cycling, which in turn imperil structural stability and long-term reliability. The incorporation of heat-shrinkable fibers, owing to their exceptional mechanical attributes and thermal contraction properties, heralds a novel and promising avenue for tackling such issues. These fibers, typically crafted from materials such as polypropylene and polyester, in this conceptualization, employ PET fibers. When incorporated into concrete, they create a three-dimensional cohesive network. Throughout the process of concrete hydration and setting, these fibers, courtesy of their intrinsic thermal contraction tendencies, seamlessly integrate with the concrete matrix, thereby notably enhancing the overall performance of the concrete. Their salient application effects encompass, yet are not confined to, the following aspects: Firstly, in the realm of crack resistance, heat-shrinkable fibers exert a notable impediment to the formation and extension of cracks. The susceptibility of high-strength marine engineering concrete to fissures under the influence of factors such as seawater corrosion and temperature fluctuations is mitigated by the introduction of these fibers, effectively elevating the concrete’s ductility and facilitating control over crack propagation. Secondly, these fibers exert a vital role in enhancing durability. The presence of chloride ions in seawater substantially impacts the corrosion of concrete rebars, and the introduction of fibers can decelerate the ingress of chloride ions, thereby retarding the internal corrosion process of concrete and enhancing structural durability. Furthermore, the addition of fibers significantly augments concrete’s toughness and load-bearing capacity, thereby bolstering overall performance. In the realm of high-strength marine concrete structures, this reinforcement effect proves crucial in withstanding the challenges posed by waves, surges, and external loads, ensuring the long-term stability of structures. However, it is pertinent to underscore that the application of heat-shrinkable fibers necessitates a comprehensive consideration of engineering design and practical construction requirements. Although fibers can markedly enhance the concrete’s properties, they do not supplant the role of conventional steel reinforcement. Consequently, in engineering practice, a judicious amalgamation of diverse reinforcement materials and methodologies is requisite to attain optimal structural performance. In summation, the application of heat-shrinkable fibers in high-strength large-volume marine concrete not only effectively augments crack resistance, durability, and overall performance but also offers valuable technological support to address the rigorous demands of marine environments, thus enhancing the sustainability and reliability of marine engineering endeavors.

Based on the characteristics of heat-shrinkable fibers developed and applied in the textile industry, the introduction of these high-modulus fibers into high-strength large- volume marine concrete has proven to be beneficial. The fibers are evenly dispersed throughout the concrete matrix, ensuring a uniform distribution by implementing surface roughening and hydrophilic modification techniques [31,32,33]. During the concrete heating phase, the high-modulus heat-shrinkable fibers undergo rearrangement and contraction stimulated by the exothermic reaction of the cement hydration. Based on the interfacial bond between the fibers and the cementitious slurry, they exert micro-prestressing stress on the concrete, thereby enhancing its tensile strength and crack resistance (Figure 6). This concept resembles the pre-tensioning method applied to concrete beams, where prestressing steel strands are tensioned before concrete pouring. Once the concrete reaches a certain strength and elastic modulus, the tension is released and the steel strands undergo elastic shrinkage. Due to the interfacial bond with the cementitious slurry, they impart prestressing stress to the concrete, improving its crack resistance and load-bearing capacity. Building upon this technical approach, researchers conducted extensive experiments and applied the newly modified heat-shrinkable fibers from the textile industry in the construction of the solid sections of the C55 main tower piers of the Huangmao Sea Cross-Sea Bridge and the C60 prestressed box girder of the Stone Beach Bridge in Guangzhou, Zengcheng. Without the incorporation of heat-shrinkable fibers [34,35,36,37], the C55 main tower piers of the Huangmao Sea Cross-Sea Bridge exhibited numerous cracks, whereas the introduction of heat-shrinkable fibers prevented crack formation. Similarly, the C60 prestressed box girder of the Stone Beach Bridge did not exhibit cracks after incorporating the heat-shrinkable fibers. These practical engineering applications demonstrate the effective crack control provided by the heat-shrinkable fibers in large-volume concrete structures. However, further research is needed to understand the mechanisms of thermally stimulated shrinkage and its coupling effect on the interfacial bond with the cementitious slurry, as well as the influence of prestressing stress and the microstructure of high-strength large-volume marine concrete. This knowledge will provide a theoretical basis for the preparation of heat-shrinkable fibers that are suitable for high-strength large-volume marine concrete.

High-strength large-volume concrete undergoes both heating and cooling stages. During the heating stage, the concrete undergoes thermal expansion, where the heat-shrinkable fibers are thermally stimulated to shrink, exerting micro-prestress on the concrete (Figure 7). In the cooling stage of high-strength large-volume concrete preparation, the shrinkage of the heat-shrinkable fibers has already occurred, whereas the concrete undergoes temperature shrinkage, autogenous shrinkage, and drying shrinkage. These shrinkage phenomena lead to a loss of the micro-prestress applied by the fibers, thereby weakening their crack resistance enhancement. Therefore, in order to enhance the crack resistance effect of the fibers [38,39,40], it is crucial to take measures to reduce concrete shrinkage and understand the mechanism by which it controls the loss of prestress in the fibers during the cooling stage of high-strength large-volume concrete preparation.

## 4. The Influence of Internal Curing Materials on Concrete Shrinkage

The investigation into the impact of internal curing materials on concrete shrinkage is a pivotal subject within the realm of concrete technology. This topic revolves around comprehensively understanding the influence of various types of internal curing materials on concrete shrinkage performance and seeks to optimize their practical application to control concrete shrinkage. In recent years, both the academic and engineering communities have exhibited an increasing interest in this subject and aimed to gain a deeper grasp of the essence of concrete shrinkage issues and potential solutions. Internal curing materials exert their influence on concrete shrinkage performance through various mechanisms. Firstly, these materials effectively retain moisture within the concrete, thereby slowing down the rate of moisture loss and consequently mitigating drying shrinkage. Secondly, the introduction of internal curing materials can decrease the evaporation rate from the concrete surface, thereby alleviating the impact of drying shrinkage. Moreover, some internal curing materials might interact with the cementitious gel in the concrete, influencing the progress of hydration reactions and thus, to some extent, controlling concrete shrinkage. Internal curing materials come in a diverse array of guises, including, but not limited to, liquid membranes, dampened paper, wet cloths, polymer films, etc. Research indicates that the selection and application methodology of these materials can yield varying degrees of influence on different types of concrete shrinkage, such as drying shrinkage and thermal shrinkage. For example, materials such as liquid membranes and wet cloths can create a sealed physical barrier, effectively slowing moisture evaporation and thereby reducing drying shrinkage. On the other hand, the introduction of polymer films and similar materials might influence hydration reactions and thus control thermal shrinkage.

In concrete shrinkage studies, measurement techniques are indispensable tools. Strain gauges, drying shrinkage tests, thermoluminescence measurements, and other techniques are extensively utilized to evaluate the influence of internal curing materials on concrete shrinkage. These techniques provide quantifiable data that aid researchers in more accurately revealing the relationship between internal curing materials and concrete shrinkage. In studies of internal curing materials, performance comparisons are often crucial. Researchers frequently compare the performance of concrete using internal curing materials with concrete cured through traditional methods. Such comparative studies not only evaluate the practical effectiveness of internal curing materials but also explore their effects on shrinkage control, strength development, and durability. In terms of practical engineering application, an increasing number of studies are examining the effects of internal curing materials in real-world projects. These studies consider not only the material’s cost, feasibility, and construction conditions but also involve long-term monitoring and assessment of the material’s performance. This contributes to guiding the selection and utilization of internal curing materials in engineering practice and provides a scientific basis for the durability and reliability of projects.

In conclusion, research on the influence of internal curing materials on concrete shrinkage is continuously advancing, offering fresh insights into more effective methods of controlling concrete shrinkage. However, certain aspects still require further exploration, such as the mechanisms behind the influence of various types of internal curing materials and the feasibility and longevity of their application in real-world projects. With increasing research in these domains, a deeper comprehension of the interaction between internal curing materials and concrete shrinkage can be attained, providing a more robust scientific basis for enhancing the design and execution of concrete engineering projects.

Research indicates that the incorporation of internal curing materials can mitigate concrete shrinkage. The commonly used internal curing materials for concrete are lightweight aggregates and superabsorbent polymers (SAP) that are primarily used due to their affordable price and abundant availability. SAP, compared with lightweight aggregates, exhibits superior desorption capacity (Figure 8). However, the release of moisture by SAP can create voids within the dense cementitious matrix, leading to a reduction in the mechanical properties and durability of concrete and rendering it unsuitable for high-strength large-volume concrete in marine engineering. On the other hand, conventional lightweight aggregates [41,42,43,44] tend to diminish the mechanical properties of concrete. Additionally, their water release mechanism during the temperature fluctuations in large-volume concrete is closely associated with their pore structure and the ambient temperature and humidity. During the heating phase of high-strength large-volume concrete, the lightweight aggregates experience a rapid temperature rise, reaching internal temperatures as high as 70–90 °C. This accelerated heating prompts the quick release of moisture by the lightweight aggregates, resulting in increased and more abundant voids in the surrounding cementitious slurry. Consequently, during the subsequent cooling phase of large-volume concrete, the water release from the lightweight aggregates diminishes, thereby reducing their effectiveness in internal curing. However, in the preparation of internal curing aggregates, it is essential to optimize their pore structure, creating highly refined interconnected pores with high water absorption and water retention capabilities. This way, during the concrete curing process, these aggregates can slowly release moisture as the concrete heats up and quickly release moisture as the concrete cools down. This controlled moisture release process not only aids in internal curing, reducing self-shrinkage and drying shrinkage caused by capillary pore negative pressure, but the porous internal curing aggregates’ insulating effect can also the lower temperature differences and cooling rates on the internal surfaces of concrete structures. This, in turn, reduces the loss of fiber-induced micro-prestress stress due to concrete shrinkage and creep during the cooling phase.

In order to address potential issues arising from internal curing materials in high-strength large-volume concrete, researchers have been exploring innovative solutions. One promising alternative approach involves the use of nanomaterials, such as nano oxides or carbon nanotubes, as internal curing agents. These nanomaterials possess extremely small dimensions, allowing them to more effectively fill the micro-voids within the concrete, and their adsorption and release of moisture can be fine-tuned by controlling particle size and surface modifications. As a result, nanomaterials can provide more stable moisture release under high-temperature conditions, contributing to the preservation of concrete’s mechanical properties and durability. Furthermore, some advanced high-performance concrete technologies have embraced the concept of self-healing materials. These materials can autonomously repair microcracks within the concrete, alleviating shrinkage concerns. These self-healing materials may include microcapsules containing healing agents that can be released when needed or micro/nanofibers that can bridge microcracks and enhance the continuity of the concrete.

In recent years, researchers have developed a spherical high-microscopic interconnected-pore internal curing aggregate with a diameter ranging from Φ1.00 to 4.75 mm. This aggregate [45,46] exhibits high strength, water storage stability, and adjustable water absorption capacity. It can partially replace the fine aggregate in high-strength large-volume concrete used in marine engineering. By serving as an internal curing agent, it effectively reduces concrete shrinkage and mitigates the micro-prestress stress loss caused by the shrinkage and creep of the aforementioned shrinkage-reducing fibers, thereby enhancing the crack resistance of the concrete. The porous internal curing aggregate also provides thermal insulation, which helps decrease temperature differentials on the surface of large-volume concrete. The transitional zone between the internal curing aggregate and the cementitious matrix forms an arching shell structure, enabling the even distribution of compressive stress and slowing down the formation and propagation of cracks. This further improves the crack resistance of high-strength large-volume concrete in marine engineering.

Methods for the preparation of high-microscopic interconnected-pore internal curing aggregates with water absorption, medium-high strength, and a predominant presence of microfine interconnected pores have been proposed and industrially produced in Yichang. A water release model for prewetted internal curing aggregates [47] in cementitious materials has been established, revealing their influence on the range of water release from internal curing in hardened paste and elucidating the formation mechanism and influencing the factors of the strengthened arch–shell interface transition zone (Figure 9, Figure 10 and Figure 11). The high-microscopic interconnected-pore internal curing aggregates can be used in the preparation of highly abrasion-resistant ultrahigh-performance concrete by partially replacing the fine aggregate. Engineering demonstrations have been conducted in the maintenance and reinforcement project of Fuhexi Bridge in Wuhan City.

This novel type of highly interconnected micro-porous internal curing aggregate brings a significant innovation to concrete technology in the field of marine engineering. Its unique attributes, including high strength, adjustable water absorption capacity, and exceptional micro-interconnected pore structure, render it an ideal choice for large-span sea-crossing bridge projects. These attributes not only significantly enhance the durability and crack resistance of concrete but also help mitigate the impact of temperature variations on the concrete surface, thereby reducing the potential for cracks and damage. The application of internal curing aggregates has been validated in engineering practice, especially in the maintenance and reinforcement project of the Fuhexi Bridge in Wuhan. By partially substituting traditional fine aggregates with these highly interconnected micro-porous internal curing aggregates, a highly abrasion-resistant ultrahigh-performance concrete was successfully developed, showcasing its feasibility and effectiveness in real-world projects. This successful case serves as a robust reference for future similar projects while also paving the way for new directions in the innovation and advancement of concrete technology in China. The introduction of this new type of internal curing aggregate enriches the array of choices for marine engineering concrete materials and promotes the sustainable development of marine engineering construction. Against the backdrop of strategies such as the “Belt and Road Initiative,” this innovation not only strengthens China’s technological prowess in the field of marine engineering but also provides robust support for regional cooperation and international exchange. Through continuous research and practical applications, we can foresee that this novel internal curing aggregate will play an increasingly significant role in future marine engineering projects, offering a reliable foundation for more durable and dependable engineering structures.

However, the mechanisms of water release during the temperature changes and their effects on the microstructure of the concrete in the ascending and descending phases are yet to be fully understood. Moreover, there is currently a lack of research on the synergistic control of microstructure and performance enhancement in high-strength large-volume concrete in marine engineering using both shrinkage-reducing fibers and internal curing aggregates [42,46,48]. The mechanisms underlying their cooperative regulation of concrete microstructure and performance enhancement need to be elucidated.

## 5. Conceptual Framework for The Application of Heat-Shrinkable Fibers and Internal Curing Aggregates in the Field of the Crack Resistance of High-Strength Large-Volume Marine Concrete

Based on the aforementioned analysis, considering the developed application of heat-shrinkable fibers and highly micro-connected-pore curing aggregates [49,50,51,52], the following concept is proposed: introducing both heat-shrinkable fibers and highly micro-connected-pore curing aggregate into high-strength large-volume marine concrete with a compressive strength exceeding C50 in order to synergistically regulate its microstructure and enhance its crack resistance and resistance to chloride salt erosion and thereby addressing the technical challenges of crack control and resistance to chloride salt erosion in high-strength large-volume marine concrete (Figure 12).

(1) Concept for synergistically enhancing concrete crack resistance: leveraging the high modulus of the heat-shrinkable fibers and their coupled effect of thermal shrinkage-induced contraction at the interface with the cementitious slurry, exerting three-dimensional micro-prestress on the concrete to improve its tensile strength and ductility, while simultaneously enhancing the mechanical properties of the arch-shell structure at the interface between the internal curing aggregate and the cementitious slurry; employing the internal curing aggregate to suppress concrete shrinkage and reduce the loss of micro-prestress caused by concrete shrinkage and creep; dispersing internal stresses by forming an arch-shell structure at the interface between the internal curing aggregate and the cementitious slurry [52,53,54,55,56]; employing porous internal curing aggregate for thermal insulation, thereby reducing temperature gradients within the concrete.

(2) Concept for synergistically enhancing concrete resistance to chloride salt erosion: utilizing the three-dimensional micro-prestress induced by the heat-shrinkable fibers to improve the inherent defects of the cementitious slurry; forming an arch-shell structure at the interface between the internal curing aggregate and the cementitious slurry to enhance its interface microstructure; synergistically improving the fiber, concrete aggregate, and internal curing aggregate interface microstructure by means of the three-dimensional micro-prestress and internal curing.

In Figure 12, it can be found that when introducing both heat-shrinkable fibers and high-performance micro-porous internal curing aggregate into high-strength large-volume marine concrete, they mutually influence the microstructural aspects of the concrete. The three-dimensional micro-prestressing stress applied by the heat-shrinkable fibers results in a denser microstructure of the cementitious slurry and its interface with concrete aggregates, thus improving the inherent defects within the cementitious slurry matrix and at the interface between the aggregates and the cementitious slurry. The high-performance micro-porous internal curing aggregate, partially replacing fine aggregates in the concrete, further enhances the internal curing effect and improves the microstructure of the cementitious slurry matrix, concrete aggregates, and the interface between heat-shrinkable fibers and the cementitious slurry. This reduction in concrete shrinkage and the loss of prestressing stress imposed by the fibers increase the concrete’s tensile strength, ultimately enhancing its crack resistance. Furthermore, the internal curing effect encourages the formation of an arching shell structure at the transition zone between spherical high-performance micro-porous internal curing aggregates and the cementitious slurry. The three-dimensional micro-prestressing stress imposed by the heat-shrinkable fibers strengthens the arching shell structure of the internal curing aggregates, further improving the mechanical properties and crack resistance of the concrete.

The microstructure of high-strength large-volume marine concrete directly influences its macroscopic performance. Understanding the mechanisms through which heat-shrinkable fibers and high-performance micro-porous internal curing aggregates jointly regulate the microstructure of the concrete is crucial for enhancing its crack resistance. Additionally, understanding the inherent relationship between microstructure formation and crack resistance improvement in concrete is key to guiding the enhancement of concrete crack resistance. Therefore, elucidating the mechanisms by which heat-shrinkable fibers and internal curing aggregates co-regulate the microstructure and crack resistance of high-strength large-volume marine concrete represents the core and pivotal scientific challenge to be addressed in this project.

In response to the technical challenges posed by high-strength marine concrete, the proposed concept combines the characteristics of thermal shrinkage fibers and highly interconnected micro-porous internal curing aggregates to achieve a synergistic enhancement of concrete performance. In terms of crack resistance, the high modulus and thermal shrinkage effects of thermal shrinkage fibers, combined with the collaborative action of internal curing aggregates, establish a three-dimensional micro-prestressing effect. This leads to an improvement in the concrete’s tensile strength and ductility while effectively reducing the occurrence of cracks. This innovative concept paves a new path for the application and technological advancement of high-strength marine concrete. By implementing this concept in practical engineering projects, critical issues such as crack control and resistance to chloride salt intrusion can be effectively addressed, thereby enhancing the reliability and durability of marine engineering structures. This not only drives innovation and development in the field of marine engineering but also offers valuable insights and guidance to similar projects both domestically and internationally. Through continuous research and practical application, we can further refine and apply this concept, providing more dependable solutions for future marine engineering construction.

## 6. Conclusions and Future Development Prospect

This paper presents a comprehensive review and analysis of the research achievements in the field of the crack resistance of high-strength marine concrete. It explores future research directions concerning the synergistic regulation of crack resistance in large-volume concrete through the utilization of thermally shrinkable fibers and internal curing aggregates. The findings are summarized as follows

(1)The current status of high-strength marine concrete has been analyzed, elucidating the developmental trends and crack control methods in this domain.(2)The mechanism of applying micro-prestress to concrete using thermally shrinkable fibers has been identified. Due to their high modulus, adjustable shrinkage rate, excellent dispersibility, and strong interface bonding with the concrete matrix, thermally shrinkable fibers can effectively regulate cracking in large-volume high-strength marine concrete.(3)The water release mechanism during the temperature rise and fall stages of large-volume concrete and the effective control of crack formation and propagation in the early curing stage through the utilization of fine interconnected internal curing aggregates have been investigated.(4)The synergistic regulation of microstructure and performance in high-strength marine concrete through the combined use of thermally shrinkable fibers and internal curing aggregates, as well as a coordinated design approach for crack resistance and resistance to chloride ion erosion based on these materials, have been proposed.

This study introduces innovative technologies and materials to tackle cracking and chloride ion erosion in high-strength marine concrete. It also has theoretical significance by bridging textile and building materials and offers insights for designing more durable large-span sea-crossing bridges.

## Figures and Tables

**Figure 1 polymers-15-03884-f001:**
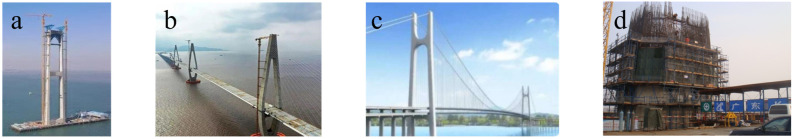
Projects of substantial magnitude in high-strength marine concrete: (**a**) The vast expanse of the Deep Zhong Ling Dingyang Bridge; (**b**) The Shanghai–Ningbo Ship Grand Passage Project; (**c**) Pictorial representation of the Lion’s Roar Bridge; (**d**) The Yellow Reed Sea Bridge.

**Figure 2 polymers-15-03884-f002:**
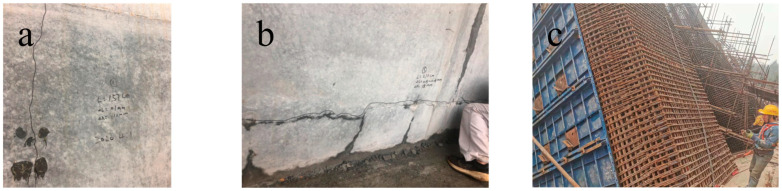
High-strength large-volume marine concrete presents challenges in the installation of water pipes and exhibits evidence of cracking: (**a**) Vertical cracking of concrete; (**b**) Horizontal cracking of concrete; (**c**) Inconvenience in the installation of cooling water pipes.

**Figure 3 polymers-15-03884-f003:**
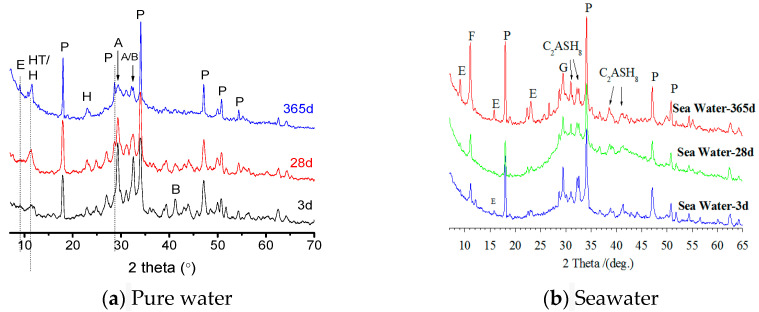
X-ray diffraction (XRD) patterns of cement–slag slurries at various maturation stages under immersion curing and seawater erosion: (**a**) Pure water; (**b**) Seawater.

**Figure 4 polymers-15-03884-f004:**
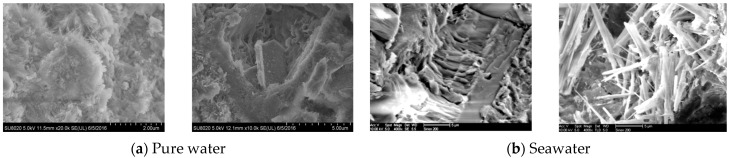
Products in cementitious slurries after 365 days of immersion curing and seawater erosion: (**a**) Pure water; (**b**) Seawater.

**Figure 5 polymers-15-03884-f005:**
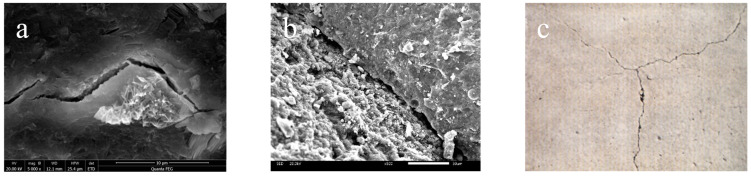
Cracks induced in large-volume concrete due to the thermal expansion of fiber reinforcement: (**a**) Microscopic cracks; (**b**) Submicroscopic cracks; (**c**) Macroscopic cracks.

**Figure 6 polymers-15-03884-f006:**
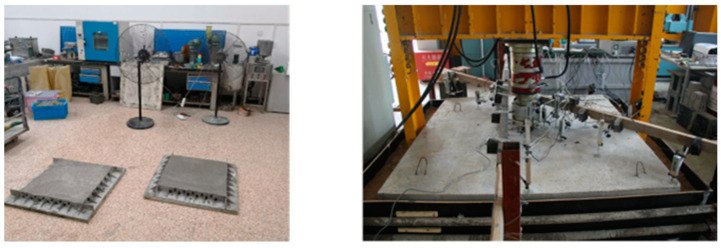
The experimental investigation of the influence of heat-shrinkable fibers on the crack resistance of concrete.

**Figure 7 polymers-15-03884-f007:**
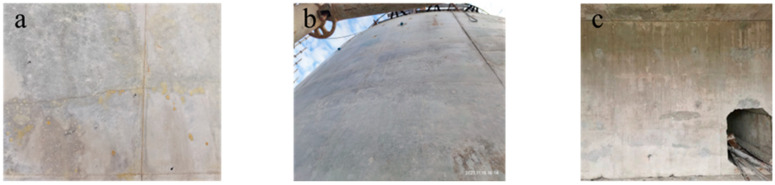
Comparison of the crack control effectiveness by incorporating heat-shrinkable fibers: (**a**) Significant fissures observed in the absence of heat-shrinkable fibers; (**b**) Solid segment of Huangmaohai main tower and Shitan Bridge; (**c**) Absence of fissures upon the incorporation of heat-shrinkable fibers.

**Figure 8 polymers-15-03884-f008:**
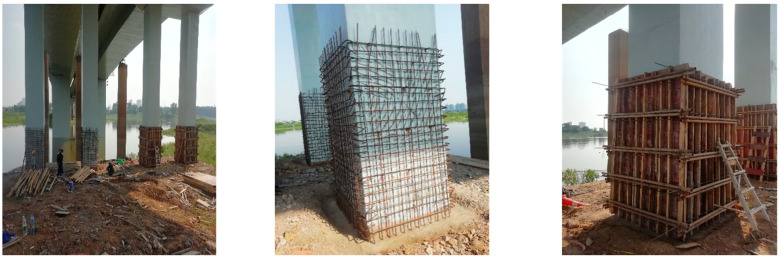
The preparation of high-strength, ultrahigh-performance concrete incorporating highly refined interconnected internal curing aggregates for application in the abrasion-resistant protection of bridge piers.

**Figure 9 polymers-15-03884-f009:**
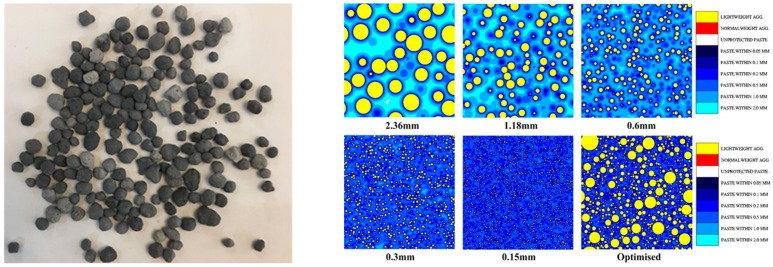
Simulation of the internal curing range of high-fired finely interconnected pores and their corresponding curing aggregates.

**Figure 10 polymers-15-03884-f010:**
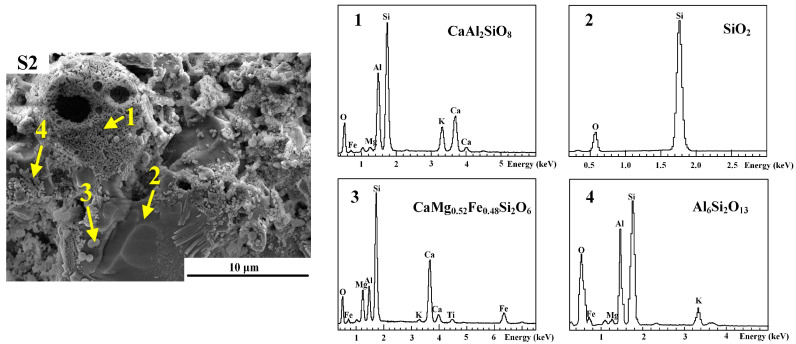
Secondary electron image and energy-dispersive X-ray spectroscopy (EDS) analysis of high-fired finely interconnected-pore internal curing aggregates at 1200 °C.

**Figure 11 polymers-15-03884-f011:**
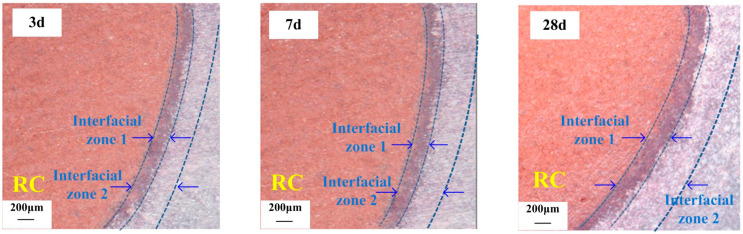
Transition zone of the cement paste–internal curing aggregate arch-shaped interface at various curing ages.

**Figure 12 polymers-15-03884-f012:**
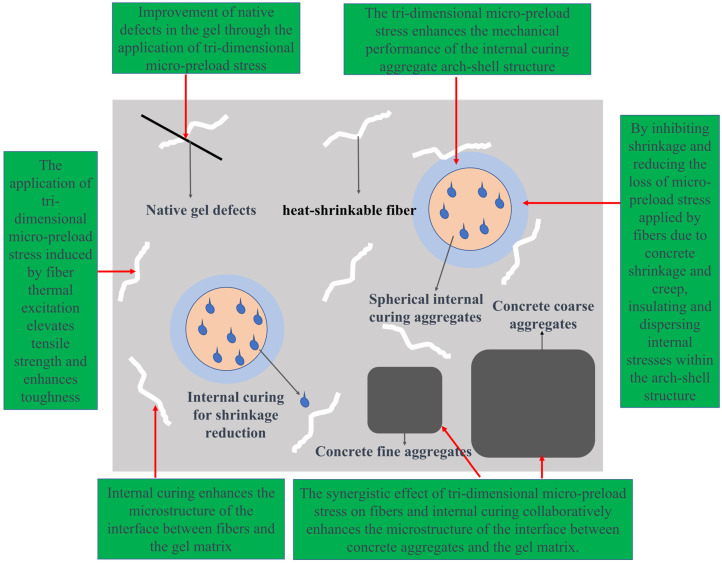
Conceptual illustration of crack control and resistance to chloride ion erosion in high-strength large-volume marine concrete.

**Table 1 polymers-15-03884-t001:** Relevant parameters of PET fiber.

Fiber Type	Diameter	Length	Strength	Melting Point
PET fibers	15 denier	51 mm	3.5 MPa	250 °C

## Data Availability

Not applicable.

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
