# Peer review of "The Application of Heat-Shrinkable Fibers and Internal Curing Aggregates in the Field of Crack Resistance of High-Strength Marine Structural Mass Concrete: A Review and Prospects"

_polymers, 2023, doi:10.3390/polym15193884_

Round 1

Reviewer 1 Report

The paper is written in great way and it requires little more effort to make better version.

I would suggest authors to address the following points to make manuscript high standard.

In abstract, line 20-21,

These fibers exploit the heat generated during concrete hydration to induce52 shrinkage 20 and apply three-dimensional micro-prestressing stress to enhance the crack resistance of the concrete.

The above sentence should be modified.

The authors have to polish abstract in terms of language and the main focus of the review.

Line 121,

The authors have to add different types rust inhibitors and their importance

Line 140,

Authors mentioned that in case of C50 constrictions, it is not possible to install cooling water pipes; further research is required to develop novel methods for controlling cracks. In this case, temperature rises to more than 70C which damages constructions due to establishment of cracks. I can see that there are few studies need to include on controlling cracks and installation of cooling pipes.

Line 169,

Expansive agents and hydration heat suppressants can reduce the temperature differential on the concrete surface, and help to prevent from cracks for above C50 constrictions. The authors have to elaborate this topic with latest articles which I have seen.

Line 495,The authors have explained conceptual frame work in a great manner. If authors include imaginary schematic figure related cracks for C50 at higher temperature, readers would understand importance of different agents to control heat effects.

Requires moderate 

Author Response

Dear reviewer

        We feel great thanks for your professional review work on our article. We have diligently perused the suggestions you have proffered, incorporating the requisite revisions into the original manuscript. The alterations have been duly appended as accompanying enclosures. We earnestly beseech your guidance for any shortcomings that may persist. Finally, we extend our heartfelt gratitude for the time and effort you have dedicated to this document. With sincere regards, we wish you good health and smooth endeavors.

Reviewer 2 Report

The authors describe the theoretical foundations and technical support for preparing crack-resistant marine structural mass concrete used in large-span bridges.

The introduction and the state of the art clearly describe the paper's topics. However, the section that studies the internal curing materials on concrete shrinkage should be enlarged to describe the subject better. The section on the conceptual framework is too short and does not give sufficient information. 

Internal Curing Materials on Concrete Shrinkage

Please improve the English language. A lot of mistyping and errors are present in the text. 

Author Response

(The authors gave the same response as above.)

Reviewer 3 Report

The authors have presented a highly intriguing approach that utilizes heat-shrinkable fibers and internal curing aggregates to enhance the crack resistance of high-strength marine structural mass concrete required for sea-crossing bridge construction. These materials employ micro-prestressing stress and internal curing effects to mitigate cracking issues and improve concrete durability. The synergistic application of heat-shrinkable fibers and internal curing aggregates in large-volume marine concrete is novel and provides valuable insights for concrete development and research. However, there are still some issues in the manuscript that need to be addressed before publication.

1. In the manuscript, please provide a detailed explanation of how heat-shrinkable fibers primarily affect the crack resistance performance of concrete, including the underlying mechanisms

2. In Table 1, relevant parameters of PET fibers should be appropriately supplemented.

3. Lines 228-230: Please explain why the tensile strength increases by 15% to 20% when steel fibers are added to the concrete.

4. Please analyze the main functions of each component in the schematic diagram of Figure 12, as well as the interactions between them.

5. Conclusion section should summarize more clearly and explicitly.

6. The manuscript contains some grammar misplacements. Please diligently address and correct them.

it is fine

Author Response

(The authors gave the same response as above.)

Round 2

Reviewer 1 Report

Publish it as it is